# Proton Therapy in the Management of Luminal Gastrointestinal Cancers: Esophagus, Stomach, and Anorectum

**DOI:** 10.3390/cancers14122877

**Published:** 2022-06-10

**Authors:** Jana M. Kobeissi, Charles B. Simone, Lara Hilal, Abraham J. Wu, Haibo Lin, Christopher H. Crane, Carla Hajj

**Affiliations:** 1Department of Radiation Oncology, School of Medicine, American University of Beirut Medical Center, Beirut 1007, Lebanon; jmk24@mail.aub.edu (J.M.K.); lh54@aub.edu.lb (L.H.); 2Department of Radiation Oncology, New York Proton Center, New York, NY 10035, USA; csimone@nyproton.com (C.B.S.II); hlin@nyproton.com (H.L.); 3Department of Radiation Oncology, Memorial Sloan Kettering Cancer Center, New York, NY 10027, USA; wua@mskcc.org (A.J.W.); cranec1@mskcc.org (C.H.C.)

**Keywords:** esophageal cancer, gastric cancer, anorectal cancer, luminal gastrointestinal cancers, proton beam therapy, pencil beam scanning, toxicity

## Abstract

**Simple Summary:**

Radiation treatment plays a major role in the management of luminal gastrointestinal cancers, mainly esophageal and anorectal cancers. There is a growing interest in the application of protons for gastrointestinal cancers, mainly owing to its dosimetric characteristics in decreasing dose to nearby organs at risk. We present here an up-to-date comprehensive review of the dosimetric and clinical literature on the use of proton therapy in the management of luminal gastrointestinal cancers.

**Abstract:**

While the role of proton therapy in gastric cancer is marginal, its role in esophageal and anorectal cancers is expanding. In esophageal cancer, protons are superior in sparing the organs at risk, as shown by multiple dosimetric studies. Literature is conflicting regarding clinical significance, but the preponderance of evidence suggests that protons yield similar or improved oncologic outcomes to photons at a decreased toxicity cost. Similarly, protons have improved sparing of the organs at risk in anorectal cancers, but clinical data is much more limited to date, and toxicity benefits have not yet been shown clinically. Large, randomized trials are currently underway for both disease sites.

## 1. Introduction

As cancer survivors are living longer, quality of life is of increasing importance both in the short term, meaning during and immediately following treatment, and in the long term during survivorship. It is thus becoming more essential to minimize or even avoid treatment-related toxicities, especially in the long term. The same radiation dose that is meant to damage the DNA of tumor cells may also damage that of nearby benign cells [1]. Photon therapy, the most commonly used radiation modality, has a dose that peaks and then exponentially declines while traversing the body, thus affecting tissues beyond where the maximum dose should lie [2]. Proton therapy, in comparison, deposits a dose at a specific depth with no exit dose and minimal collateral irradiation to the nearby organs at risk, which could lead to less acute and long-term complications [2]. Accelerated proton beams are narrow and require widening, both longitudinally and laterally, to be able to cover three-dimensional targets appropriately. This can be done with either scattering or scanning techniques. Passive scattering (PS) modalities were adopted first, and they relied on electro-mechanical methods for beam widening. Later, scanning techniques, which rely on magnetic scanning of proton beamlets, were introduced. Uniform scanning (US) served as a transition modality before being replaced by pencil beam scanning (PBS) modalities instead. The latter is superior to PS and US in dose conformity and has allowed for the application of intensity modulated proton therapy (IMPT) [2]. 

Radiation therapy plays a key role in the treatment of different gastrointestinal cancers [3], and the application of proton therapy has been proposed to decrease toxicity for these tumors [4]. While dosimetric studies are abundant, clinical data are still relatively scarce. Several review articles have been published so far, the last comprehensive one being in 2018 [5]. With more and more centers adopting proton therapy, the latter has been the subject of active research, especially over the past few years [6]. We present in this paper an up-to-date, comprehensive review of the literature on proton radiation treatment for esophageal, gastric, and anorectal cancers.

## 2. Methods

In this narrative review, we relied on PubMed to find past literature. The search results were scanned by title and abstract, and relevant articles were included with no specific exclusion criteria. To be comprehensive, we also scanned the references and the list of “cited by” papers for each of the relevant articles. As for the prospective studies, we relied on www.clinicaltrials.gov (accessed on 1 May 2022). The search results were again scanned to select for relevant, ongoing trials. The findings of our search were then narrated in the review paper as per the type of study (dosimetric, clinical, or prospective trial) and the topics covered.

## 3. Esophageal Cancer

Esophageal cancer accounts for around 1% of all new cancer cases in the United States, with a current estimated 5-year relative survival rate of 19.9% [7]. As per the National Comprehensive Cancer Network (NCCN) Guidelines, management consists of surgical resection and/or chemoradiation (neoadjuvant or definitive), with 3D conformal radiation (3D-CRT) or intensity-modulated radiation therapy (IMRT). When the organs at risk (OARs) cannot be sufficiently spared, however, proton therapy is suggested as a preferred modality [8].

### 3.1. Dosimetric Data

In a comparative study, Isacsson et al. showed that protons may offer higher tumor control probability than photons when assuming the same level of normal tissue complication probability [9]. Later, Zhang et al. demonstrated a significant reduction in mean lung dose via a three-beam passive scattering proton plan [10]. While the latter did not result in a significantly reduced mean dose to the heart, subsequent studies using alternative planning techniques did achieve reduced heart doses [11,12]. When comparing proton and IMRT plans for patients with mid and distal tumors, Shiraishi et al. demonstrated a decreased dose received by the whole heart as well as several cardiac substructures [13]. For thoracic tumors, especially for mid and distal esophageal tumors, the dose to the heart can be further decreased when PBS plans are used as opposed to PS plans [14]. 

Improved sparing of the OARs was also maintained, even with dose escalation to the target volume and at different tumor locations. In one study, boosting the distally located gross tumor volume to 65.8 Gy via IMPT still reduced the dose received by the heart, lungs, and liver when compared to IMRT [15]. Warren et al. suggested a different comparison, one they considered to be fairer, between PBS and volumetric modulated arc therapy (VMAT) [16,17]. With a boosted planning target volume up to 62.5 Gy for mid-esophageal tumors, the heart, lungs, and bone marrow were shown to receive less irradiation with proton therapy. Of note, boost coverage was found to be less robust when setup errors were considered. Other comparisons between proton beams and VMAT also confirmed dose reductions to the heart, lungs, abdominal structures [18,19], and vertebral bone marrow [20] with protons. However, of note is that dose escalation for esophageal cancer has been shown to be of no benefit in randomized trials [21], making the use of proton therapy for dose escalation questionable. 

### 3.2. Clinical Data

The first case report of clinical data was published by Shibuya et al. in 1989 [22]. It was a case of an esophageal tumor that had invaded the aorta and the left main bronchus. After neoadjuvant radiation therapy, surgical resection yielded complete pathologic response. 

Over the next two decades, investigators from Japan’s University of Tsukuba reported on their experience with proton therapy for esophageal cancer. Most patients in those studies had squamous cell carcinoma (SCC) and received higher doses of radiation without chemotherapy. One study reported on the outcomes of 15 patients with esophageal cancer who received either proton beam therapy (PBT) alone or photon radiation therapy (XRT) with a proton boost to an escalated mean total dose of 80.4 Gy [23]. Complete responses were noted in all 15 patients, with no local recurrence observed in those with superficial tumors (*n* = 6). Out of the 9 patients with advanced tumors, however, 3 relapsed. Another study reported a 10-year disease specific survival rate of 87.5% and 38.1% for superficial and advanced lesions, respectively. This study included 30 patients who were treated with PBT ± XRT to a mean dose of 77.7–80.7 Gy [24]. A similar pattern of stage-dependent survival was shown by Sugahara et al., with a 5-year disease specific survival rate of 95% and 33% for T1 and T2-4 lesions, respectively [25]. A comparable study reported no grade 3 (G3) cardiopulmonary toxicities, but a higher risk of esophageal ulcers was observed with doses >80 GyE [26].

More recent studies from Japan reported the outcomes of concurrent chemo-proton radiation for SCC. Ishikawa et al. followed up 67 patients, receiving 60 GyE in 30 fractions, for a median duration of 49 months [27]. At 50 GyE, they were re-assessed via endoscopy, and those with suspected residual tumors received an additional 2–10 GyE boost. The resultant 4-year OS rates were 96%, 73%, and 40% for stages I, II, and III, respectively. Another multi-institutional study of 202 patients, almost half of whom had stage III-IV disease, reported a 5-year OS rate of 56.3% [28]. Even though patients received a median total dose as high as 87.2 GyE, the toxicity profile was favorable. Grade 3 esophageal ulcers were observed in just 4% of the patients. 

The first study published from the Western world, where adenocarcinoma is more prevalent, was by Lin et al. in 2012 [29]. Sixty-two patients with stage Ib-IV esophageal cancer received concurrent chemo-proton radiation at a median dose of 50.4 Gy. Of those patients who eventually underwent surgery, 28% had a complete pathologic response (pCR). A similar rate of pCR (25%) was reported by Zeng et al. whose patients all had R0 resection [30]. In another study, pathologic complete response rate was retrospectively found to be similar between a cohort of proton- versus photon-treated patients, despite the proton cohort having more advanced disease [31]. In another study, 19 patients received CRT with IMPT, resulting in a complete clinical response rate of 84.2% and a survival rate of 100% at 1 year [32].

Besides that, proton therapy was shown to decrease the rate of complications before and after surgical resection. Wang et al. demonstrated that advanced techniques (PBT/IMRT) led to a significant reduction in pulmonary and gastrointestinal complications compared to 3D-CRT within 30 days after surgery [33]. This was also true of cardiac events and wound complications [34], as well as hematologic [35] complications. Proton therapy also led to significantly lower rates of grade 4 lymphopenia, whether in the neoadjuvant [36] or definitive settings [37,38].

Esophageal proton reirradiation was also assessed. Fernandes et al. were able to generate 13 feasible plans for 14 esophageal cancer patients who had a history of prior thoracic radiation, amounting to a mean cumulative dose of 109.8 Gy [39]. The median OS was 14 months. Grade 3+ non-hematologic toxicities, whether acute or late, developed in 5 patients. Of note, all patients but one received protons via the passive scattering technique. Another study generated PBS proton plans for 17 patients with a history of head and neck and thoracic radiation to a median cumulative dose of 104.7 Gy [40]. The resultant median OS was 19.5 months, with acute and late G3+ toxicity noted in 2 and 5 patients, respectively. Given the high-risk profile of this population and the scarcity of other options available, the aforementioned toxicity rates were deemed by the investigators to be acceptable. 

A limited but increasing number of studies have compared PBT and XRT head-to-head. One retrospective analysis evaluated 343 patients with esophageal cancer who either received PBT or IMRT [41]. Protons significantly improved the 5-year OS and progression free survival (PFS) rates in those with stage III cancer (OS: 34.6% vs. 25%; PFS: 33.5% vs. 13.2%; *p* < 0.05). No differences in toxicity profiles were noted. In another study, patients receiving IMPT (*n* = 32) or IMRT (*n* = 32) concurrently with chemotherapy had similar oncologic and toxicity outcomes, including 1-year OS (74% vs. 71%, *p* = 0.62), PFS (71% vs. 45%, *p* = 0.15), and treatment-related Grade 3 toxicity (16% vs. 9%, *p* = 0.71) rates [42]. Suh et al. showed no significant differences in survival or toxicity between protons and IMRT/3D-CRT [43]. In terms of quality of life (QOL), Garant et al. compared Functional Assessment of Cancer Therapy-Esophagus questionnaire changes and noted a lesser decline in QOL with PBT vs. XRT over the course of treatment (−12.7 vs. −20.6, *p* = 0.026) [44].

The highest level of evidence to date is the phase IIb randomized controlled trial (RCT) at MD Anderson Cancer Center [45]. Patients with locally advanced esophageal cancer received neoadjuvant or definitive concurrent chemoradiation (CCRT) either with PBT (*n* = 46) or IMRT (*n* = 61) at a prescription dose of 50.4 Gy in 28 daily fractions. The 3-year OS and PFS rates were similar between the two groups (OS: 51.2% vs. 50.8%, PFS: 44.5%, *p* > 0.05). However, the randomized arms significantly differed in the total toxicity burden (TTB), which was the primary endpoint of the trial. Up to 12 months after randomization, the TTB was 2.3 times higher in the IMRT arm across all patients, and postoperative complications were 7.6 times higher in the IMRT arm among patients receiving trimodality therapy. This study suggested that protons could deliver the same efficacy at a lower toxicity cost. This study was also the first randomized proton versus photon trial showing a benefit to proton therapy in the study primary endpoint [46].

While it may seem contradictory that the above RCT noted a decrease in toxicity after proton therapy compared to the “toxicity-negative” studies by Xi et al. [41] and Bhangoo et al. [42], such a difference may be linked to what each study labeled and measured under toxicity. The retrospective studies evaluated treatment-related toxicities as per the Common Terminology Criteria for Adverse Events (CTCAE). The RCT, on the other hand, calculated a composite toxicity score, i.e., the TTB, which incorporated the total patient experience, taking into account both severity-weighted adverse events and their accumulation over 52 weeks, as well as postoperative complications. In fact, the decrease in the latter is what mainly drove the decrease in the TTB. The researchers noted no statistically apparent difference between the individual adverse events, which they attributed to the low rates of any single event. The decrease in the postoperative complications comes in line with previous literature, as mentioned above [33,34,35].

### 3.3. Prospective Trials

The largest current trial comparing PBT to IMRT is the NRG-GI006 [47]. It is a multi-institutional, randomized, phase III trial that will primarily assess the OS and occurrence of G3+ cardiopulmonary adverse events. Another randomized, phase III study (PROTECT trial) will be comparing the incidence of pulmonary complications after proton or photon therapy [48].

Other smaller trials are also underway. Each includes a single arm of patients receiving proton therapy to assess patient reported outcomes and toxicity profiles [49], OS [50], or adverse events after dose escalation [51].

Figure 1 summarizes the clinical studies on the use of proton therapy in the treatment of esophageal cancer, stratified by their special focus and the time of publication, and Table 1 summarizes comparative studies of protons versus photons for the treatment of esophageal cancers.

## 4. Gastric Cancer

Gastric cancer makes up around 1.5% of all new cancers annually in the US [52], with an estimated 5-year OS rate of 32.4% [53]. It is much more common in other areas of the world, especially East Asia, where it remains one of the leading causes of cancer death [52]. 

The standard of care for gastric cancer includes chemotherapy and surgical resection, which is sometimes followed by chemoradiation treatment (CRT). The role of the latter has been established by the INT0116 trial, which showed improved OS and relapse free survival for adjuvant CRT compared to observation [54]. For this purpose, the NCCN Guidelines recommend 3D-CRT or IMRT, with no clear role for protons [55].

### 4.1. Dosimetric Data

Dionisi et al. performed the first dosimetric study, comparing plans for adjuvant chemoradiotherapy using proton (double scattering-uniform scanning technique) versus photon radiation to a prescription dose of 45 ± 9 Gy [56]. With proton therapy, the OARs were significantly less exposed to low-medium dose range, with robust, yet less homogeneous, plans than those with photons.

Later, Mondlane et al. compared plans using protons (PBS with single-field uniform dose [SFUD]) versus photons (VMAT) in 8 patients receiving adjuvant chemoradiotherapy, with a prescription dose of 45 Gy [57]. The SFUD plans resulted in comparable or lower doses to the OARs, especially the kidneys, liver, and spinal cord. When water-filled or air-filled CT scans were considered, however, the SFUD plans proved to be less robust with changes in density. Mondlane et al. carried out another dosimetric study with the same population [58]. This time, the plans were first optimized after replacing the air cavities with a water equivalent material via the density override approach (SFUD_opt_). The plans were then recalculated based on the original air-filled CT scans (SFUD_ver_). The two sets of plans were similar, with insignificant dosimetric changes. While SFUD exposed the kidneys, liver, and spinal cord to lesser doses than VMAT, the estimated normal tissue complication probability was only reduced for the left kidney (0% for all 8 patients with SFUD_opt_ vs. 0% in just 3 patients with VMAT).

### 4.2. Clinical Data

Clinical data on the usage of proton therapy in gastric cancer are scarce. The only available evidence is case reports from the University of Tsukuba from over 30 years ago [59,60]. 

In 1990, Koyama et al. reported the case of a 72-year-old man with advanced gastric cancer, inoperable due to severe emphysema [59]. He was offered definitive chemoradiation, with the proton modality and a dose up to 61 Gy. As a result, the tumor regressed and underwent necrosis, while the surrounding normal tissue architecture was spared. Similar cases were reported by Shibuya et al. in 1991 [60], who treated two men, aged 85 and 70 years, with early, inoperable gastric cancer. They received definitive, dose escalated PBT (83–86 Gy), and follow-up endoscopy showed persistent gastric ulcers with no tumor cells. Since then, no further clinical data have been reported, and no trials are currently underway.

## 5. Anorectal Cancer

Rectal cancer is the 7th most common cancer worldwide [61], with around 44,850 new cases expected for 2022 in the US alone [62]. Anal cancer, on the other hand, is much less common, with an estimated incidence of 9,440 cases [63]. The 5-year relative survival rates for rectal and anal cancers are 67% [64] and 69% [65], respectively. 

In the earliest stages, rectal adenocarcinoma can be excised upfront. In more advanced cases, combined modality treatment is preferred, including neoadjuvant CCRT, chemotherapy (adjuvant and/or neoadjuvant), and surgery [66]. Non-metastatic anal SCC is primarily treated with CCRT, with surgery reserved for local recurrence or persistence of the tumor [67].

### 5.1. Dosimetric Data

In 1992, Tatsuzaki et al. generated different plans for a single case of rectal adenocarcinoma [68]. The proton plans spared the small bowel, bladder, and femoral heads/necks to a greater degree, thus allowing a higher dose to be directed to the tumor. Isacsson et al. showed a similar OAR sparing profile [69]. They also showed that the tumor control probability is higher with protons for the same normal tissue complication probability (NTCP 5%).

Later studies compared proton plans with more advanced photon techniques. One study compared passively scattered protons to IMRT and 3D-CRT [70]. Although all plans had comparable dose homogeneity, IMRT/3D-CRT covered the PTV to a slightly better degree, while PBT significantly spared the OARs more. For example, the small bowel volume exposed to 15 Gy was 90 cc, 138 cc, and 157 cc for PBT, IMRT, and 3D-CRT, respectively (*p* < 0.05) [70]. Another comparative study between passively scattered protons and VMAT/3D-CRT demonstrated a numeric reduction in the doses received by the bowels and the bladder, but the decrease was not statistically significant [71]. 

Plans with scanning techniques were shown to be feasible as well. In comparison to RapidArc, IMRT, and 3D-CRT, protons lowered the doses to the bladder, small bowel, and testes, but not to the anal canal [72]. In another study, 3-field uniform scanning plans significantly spared the pelvic bone marrow and small bowel more so than IMRT or 3D-CRT [73]. Given the confounding hematologic toxicity of CCRT, sparing the bone marrow is particularly significant and may obviate the need for treatment interruptions.

Kronborg et al. also investigated the bone sparing capacity of pencil beam proton therapy, while also assessing the risk of pelvic insufficiency fractures [74]. Compared to VMAT and IMRT, protons showed better sparing of the pelvic, sacral, and sacroiliac bones, especially for the low dose volumes (V20–30 Gy). On follow-up imaging with a 3-year pelvic MRI, 9 out of 27 patients were found to have pelvic insufficiency fractures, which correlated with higher doses received by their pelvic bones. 

Other studies considered special cases of rectal cancer. Radu et al. compared PBT and IMRT plans for locally very advanced rectal tumors, with the former plans showing improved OAR sparing yet less robustness [75]. Other groups of researchers looked at the possibility of pelvic reirradiation. One study included 15 patients with pelvic recurrences of their rectal cancers [76]. IMPT and VMAT plans were compared, with adequate target coverage by both. However, IMPT reduced the mean dose received by all OARs, particularly the bowels, bladder, and sacrum. In another multi-institutional study, pelvic re-irradiation with SBRT also showed superior OAR sparing [77].

As for anal SCC, Wo et al. compared PBS and IMRT [78]. With equivalent target coverage, PBS proved to spare the bowels, bladder, femoral heads, and iliac crests to a greater degree across different dose ranges. Other studies confirmed better OAR sparing with PBS, when compared to IMRT [79,80] or VMAT [81,82,83]. For example, Meier et al. demonstrated that IMPT significantly decreased the mean dose to the bone marrow when compared to VMAT (17.42 Gy vs. 30.76 Gy, *p* < 0.0001), and this translated into a predicted decrease in Grade 3+ hematologic events from 40% to <5% as per the NTCP model for bone marrow toxicity [81].

### 5.2. Clinical Data

In 1977, Suit et al. reported a case series of patients with different cancer types, including anorectal, who received PBT. Patients tolerated well both neoadjuvant and adjuvant radiation with a proton boost [84]. In 1982, Suit et al. published an expanded series of 317 patients, 14 of whom had anorectal cancer [85]. Protons were either given for the entire course or as a boost after photons, with the boost often being delayed due to perineal pain. The 3 patients who had local control and survived beyond 2 years had relatively shorter treatment times (67, 77, and 86 days) compared to the overall group (range: 67–131 days), thus highlighting the importance of minimizing interruptions. 

More than 30 years later, Lee et al. reported on the outcomes of 67 patients with locally recurrent rectal cancer (LRRC), 4 of whom received protons as part of CCRT [86]. It was noted that higher doses could be prescribed to target volumes, even with nearby OARs. However, with the small sample size, no subgroup analysis was made to evaluate the oncologic outcomes of PBT in particular.

Proton therapy to LRRC has become a topic of interest, with several case reports and retrospective studies published. In one study, 13 patients received 70 GyE in 25 fractions, resulting in a local control rate of 46% and only one G3/4 toxicity [87]. Another study retrospectively evaluated the outcomes of 12 patients who received PBT to LRRC [88]. With a median follow-up of 42.9 months, the 3-year PFS and OS rates were 12.1% and 71.3%, respectively, with no G 2+ acute or late toxicities noted.

Reirradiation is also a topic of interest in anorectal cancer [89]. One prospective study followed 7 patients with recurrent rectal tumors who were reirradiated with protons to a total sum dose of 109.8 Gy (RBE). The majority of patients (5/7) had both metabolic (reduction in SUV on positron emission tomography scans) and anatomic partial responses, and all 6 symptomatic patients had improvement in their symptoms, with 3 of them having complete resolution of pain [90]. Another study considered patients with either recurrent or de novo rectal cancer [91]. After pelvic re-irradiation with standard or hyper-fractionation, the 1-year local progression and survival rates were 52.3% and 77.2% for recurrent tumors versus 0% and 100% for de novo tumors, respectively. Grade 3+ toxicities occurred acutely in 10.7% and late in 13.3%, with one patient sustaining a Grade 5 pre-sacral hemorrhage. Moningi et al. also evaluated the feasibility of hyper-fractionation for recurrent pelvic tumors, with 15 patients receiving 39–45 Gy (RBE) in twice daily 1.5 Gy fractions [92]. The resultant 1-year and 2-year PFS rates were modest at 58.7% and 47%, respectively, with an OS rate of 67%. There were 1 acute and 2 late G3 adverse events, with no reported G 4 toxicity.

Jeans and colleagues applied neoadjuvant short-course proton therapy [93]. Eleven patients with stage IIa-IVb rectal cancer received PBS-PT to 25 Gy in 5 fractions, followed by total meso-rectal excision. With a median postoperative follow-up of 10.5 months, all patients were alive, and none had local failure. Treatment was very well tolerated with no G 2+ dermatologic, gastrointestinal, or genitourinary toxicities. 

The feasibility of definitive chemo-proton therapy for anal cancer was first evaluated by Wo et al. [94]. Twenty-five patients with anal SCC received PBS-PBT concurrent with 5-FU and mitomycin-C, with resultant 2-year colostomy-free survival, PFS, and OS rates of 78%, 80%, and 80%, respectively. The rate of G3 radiation dermatitis was 24%, thus fulfilling the study’s predetermined definition of feasibility (<48%). Nonetheless, 40% required breaks from treatment due to toxicity, most commonly neutropenia. The authors noted similar rates of G2+ toxicities to those of the RTOG 05-29 trial, questioning whether protons offer a toxicity benefit compared to IMRT [94]. Another study retrospectively reviewed the charts of 39 patients with anal SCC, who received PBS-IMPT ± concomitant cisplatin (CDDP) with 5-FU or capecitabine [95]. The 2-year relapse free survival, colostomy free survival, and OS rates were 93.8%, 91.0%, and 94.2%, respectively. Even though 9 patients (23%) required treatment breaks, only 3 were due to toxicity. A lower rate of hematologic toxicity was reported, potentially due to replacing mitomycin-C with CDDP.

The only comparative study so far is one by Mohiuddin and colleagues [96]. In this retrospective, multi-institutional comparison, 208 patients with anal SCC either received IMPT (*n* = 58) or IMRT (*n* = 150). The unadjusted 2-year locoregional relapse free survival was similar between the 2 groups (IMPT: 91%, IMRT: 88%, *p* = 0.49). Even after a propensity score-weighted analysis, the 2-year PFS was not significantly different (HR: 0.6; 95% CI, 0.4–1.1). While IMPT was dosimetrically superior to IMRT in terms of OAR sparing, the 2 groups were similar in the rates of Grade 3+ acute toxicities (68% vs. 67%, *p* = 0.96), G 3+ late toxicities (3.5% vs. 1.8%, *p* = 0.88), and hospitalization (40% vs. 33%, *p* = 0.34).

Figure 2 summarizes the clinical studies on the use of proton therapy in the treatment of anorectal cancer, stratified by their special focus and the time of publication.

### 5.3. Prospective Trials

Proton therapy for anorectal cancer is gaining more interest. One Swedish trial (PRORECT) will be randomizing approximately 254 rectal cancer patients to a proton- or photon-based neoadjuvant, short-course radiation regimen of 25 Gy over 5 fractions [97]. A second Swedish trial (SWANCA) will be randomizing 100 patients with anal SCC to receive either IMPT or photon therapy [98]. Both trials will primarily assess acute toxicities.

Other prospective, single assignment group studies are also underway. The University of Aarhus will be evaluating pelvic proton reirradiation for recurrent rectal or anal cancer in the ReRad II and ReRad III trials, respectively [99,100]. The RECCPT and IMPARC trials will be assessing local control [101] and determining the maximum tolerated dose [102] in proton reirradiation of recurrent rectal cancer. Other phase 2 trials will be looking into the safety of concurrent chemo-proton therapy for anal cancer [103,104].

Table 2 summarizes comparative studies of protons versus photons for the treatment of anorectal cancers.

## 6. Conclusions

While evidence remains scarce for gastric cancer, a growing body of evidence supports the dosimetric advantage of protons over photons in the management of esophageal and anorectal tumors. However, further studies are needed to elucidate the clinical implications of this dosimetric benefit. Early evidence supporting a toxicity reduction for proton therapy is encouraging, especially for esophageal cancer. With randomized trials currently underway, oncologists will have better answers in the upcoming few years on the potential benefits of this advanced modality. We suggest future clinical trials to incorporate not just acute treatment-related toxicities, but also sub-acute and long-term toxicities. As shown by Lin et al.’s RCT, considering a patient’s total experience may reveal more about toxicity than counting individual adverse events [45]. Furthermore, recent studies on bowel, urinary, and sexual function patient-reported outcomes [105,106] in long term survivors of anal cancer after concurrent chemoradiotherapy highlight the importance of considering long-term toxicities, validated patient-reported outcomes, and quality of life.

Another potential area for further research is using a model-based approach to select patients for whom proton therapy is most beneficial. A clinical implementation of such model-based selection was applied to head and neck cancer patients in the Netherlands [107]. In the selection process, the difference in NTCP between VMAT and proton (best-case scenario) plans was calculated. If this difference (ΔNTCP) exceeded a predetermined threshold, a robust-IMPT plan was created instead, and the patient was chosen to receive proton therapy. Such systematic selection of patients may prove to be helpful in gastrointestinal cancers and highlight the factors that predispose patients to increased toxicity from radiation, thus supporting the role of proton radiation therapy in selected patients.

## Figures and Tables

**Figure 1 cancers-14-02877-f001:**
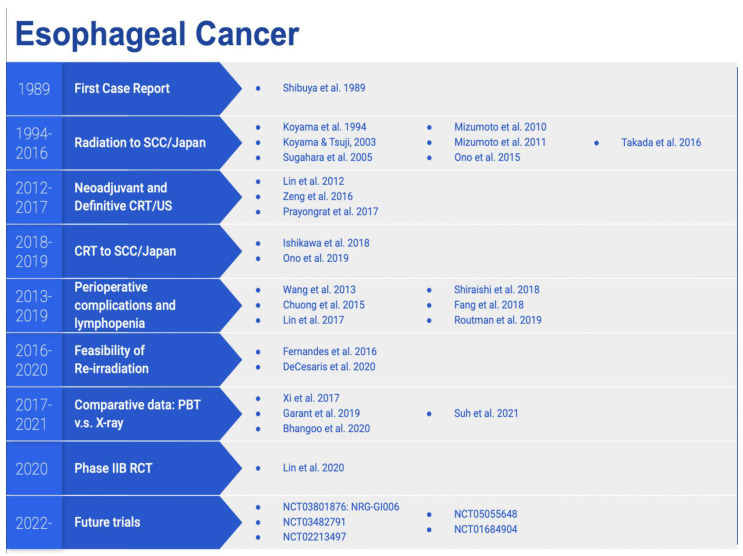
Studies on proton therapy for the treatment of esophageal cancer, stratified by their special focus and time of publication.

**Figure 2 cancers-14-02877-f002:**
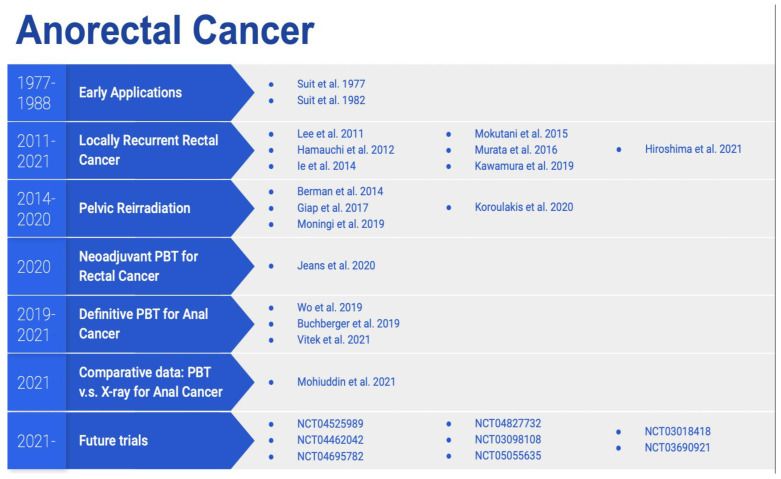
Studies on proton therapy for the treatment of anorectal cancer, stratified by their special focus and time of publication.

**Table 1 cancers-14-02877-t001:** Studies summarizing outcomes and toxicities associated with the use of proton versus photon radiation therapy for the treatment of esophageal cancer.

Authors	Year	Study Design	*n*	Comparison	OS	PFS	Toxicity
Esophageal Cancer							
Xi et al. [41]	2017	Retrospective	343		5-yr	5-yr	Grade 3/4
				PBT (1)	34.6%	33.5%	37.9%
				IMRT	25.0%	13.2%	45.0%
				*p*-value	0.038	0.005	0.192
Bhangoo et al. [42]	2020	Retrospective	64		1-yr	1-yr	Grade 3
				IMPT	74%	71%	16%
				IMRT	71%	45%	9%
				*p*-value	0.62	0.15	0.71
Lin et al. [45]	2020	Randomized phase IIB trial	107		3-yr	3-yr	TTB (2)
				PBT	51.2%	44.5%	
				IMRT	50.8%	44.5%	2.3 times higher in the IMRT arm across all patients
				*p*-value	0.60	0.70	
Suh et al. [43]	2021	Retrospective	77		5-yr	5-yr	
				PBT	64.9%	56.5%	NA
				3D-CRT or IMRT			NA
				*p*-value	0.52	0.72	
NRG-GI006 NCT03801876 [47]	2032 (est.) (3)	Randomized phase III trial	300 (est.)		Up to 8 years In progress		
				PBT			
				IMRT			
PROTECT NCT05055648 [48]	2029 (est.)	Randomized phase III trial	396 (est.)		Up to 5 years In progress		
				Proton			
				Photon			

(1) reported rates correspond to only stage III disease. (2): TTB: total toxicity burden, (3): est: estimated.

**Table 2 cancers-14-02877-t002:** Studies summarizing outcomes and toxicities associated with the use of proton versus photon radiation therapy for the treatment of anorectal cancers.

Authors	Year	Study Design	*n*	Comparison	OS	PFS	Toxicity
Anorectal Cancers							
Mohiuddin et al. [96]	2021	Retrospective	208 (1)		2-yr LRRFS (2)	2-yr PFS (3)	Grade 3+ Acute
				IMPT	91%	HR: 0.6	67%
				IMRT	88%		68%
				*p*-value	0.49	N.S.	0.96
PRORECT NCT04525989 [97]	2028 (est.)	Randomized phase II trial	254 (est.) (4)		Up to 5 years In progress		
				Photon			
				Photon			
SWANCA NCT04462042 [98]	2030 (est.)	Randomized phase II trial	100 (est.) (5)		Up to 5 years In progress		
				Proton			
				Photon			

(1,5): Anal SCC, (2): LRRFS: locoregional relapse free survival, (3): PFS: propensity score weighted progression free survival, (4): rectal cancer.

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
