# Peer review of "Proton Therapy in the Management of Luminal Gastrointestinal Cancers: Esophagus, Stomach, and Anorectum"

_cancers, 2022, doi:10.3390/cancers14122877_

Round 1
Reviewer 1 Report
The authors have written a comprehensive historical and narrative review of finished and ongoing planning and clinical studies in the field of proton therapy for GE tumors. The the article well-organized and the sections are well-developed. The authors did a good job of synthesizing the literature. The article is well-written and easy to understand. The methodology by which papers were selected and included is however less clearly explained.
Minor points:
1. the authors should elaborate more on the toxicity 'negative' trials bij Xi et al 2017 and Bhangoo et al 2020; is there something to say as to why no lower toxicity was observed in these trial in comparison to the randomized phase IIB trial?
2. In their conclusion, I encourage the authors to suggest possible endpoints for future clinical trials (acute vs late? specific complications vs total toxicity?)
3. In their conclusion, I encourage the authors to mention the model-based approach used in the Netherlands for example in H&N tumors and elaborate on possible application of this approach in oesophageal cancer DOI:https://doi.org/10.1016/j.radonc.2020.07.056
Author Response
Thank you for your comments. Kindly find below a point-by-point response.
Point 1: The methodology by which papers were selected and included is however less clearly explained.
Response 1: In this narrative review, we relied on PubMed to find past literature. The search results were scanned by title and abstract, and relevant articles were included with no specific exclusion criteria. To be comprehensive, we also scanned the references and the list of “cited by” papers for each of the selected articles. As for the prospective studies, we relied on www.clinicaltrials.gov. The search results were again scanned to select for relevant, ongoing trials. The findings of our search were then narrated in the review paper as per the type of study (dosimetric, clinical, or prospective trial) and the topics covered. This was added in a separate section of the paper, called “Methods.”
Point 2: The authors should elaborate more on the toxicity 'negative' trials by Xi et al 2017 and Bhangoo et al 2020; is there something to say as to why no lower toxicity was observed in these trial in comparison to the randomized phase IIB trial?
Response 2: In elaborating more on toxicity, we added the following under “clinical data” of Esophageal cancer:
While it may seem contradictory that the above RCT noted a decrease in toxicity after proton therapy compared to the “toxicity-negative” studies by Xi et al. and Bhangoo et al., such difference may be linked to what each study labeled and measured under toxicity. The retrospective studies evaluated treatment-related toxicities as per the Common Terminology Criteria for Adverse Events (CTCAE). The RCT, on the other hand, calculated a composite toxicity score, i.e. the total toxicity burden (TTB), which incorporated the total patient experience, taking into account both severity-weighted adverse events and their accumulation over 52 weeks as well as postoperative complications. In fact, the decrease in the latter is what mainly drove the decrease in the TTB. The researchers noted no statistically apparent difference between the individual adverse events, which they attributed to the low rates of any single event. The decrease in the postoperative complications comes in line with previous literature as mentioned above.
Point 3: In their conclusion, I encourage the authors to suggest possible endpoints for future clinical trials (acute vs late? specific complications vs total toxicity?)
Response 3: We added the following to the conclusion:
We suggest future clinical trials to incorporate not just acute treatment-related toxicities, but also late onset ones as well as complications that may arise within the duration of treatment. As shown by Lin et al.’s RCT, considering a patient’s total experience may reveal more about toxicity than counting individual adverse events [Lin et al. 2020]. Furthermore, recent studies on bowel, urinary, and sexual function patient-reported outcomes [De Brian et al. 2022, Corrigan et al. 2022] in long-term survivors of anal cancer after concurrent chemoradiotherapy highlight the importance of considering long-term toxicities, validated patient-reported outcomes, and quality of life.
Point 4: In their conclusion, I encourage the authors to mention the model-based approach used in the Netherlands for example in H&N tumors and elaborate on possible application of this approach in oesophageal cancer DOI:https://doi.org/10.1016/j.radonc.2020.07.056
Response 4: We added the following to the conclusion:
Another potential area for further research is using a model-based approach to select patients for whom proton therapy is most beneficial. A clinical implementation of such model-based selection was applied to head and neck cancer patients in the Netherlands (Tambas et al. 2020). In the selection process, the difference in NTCP between VMAT and proton (best-case scenario) plans was calculated. If this difference (ΔNTCP) exceeded a predetermined threshold, a robust-IMPT plan was created instead, and the patient was chosen to receive proton therapy. Such systematic selection of patients may prove to be helpful in gastrointestinal cancers, particularly esophageal ones, and highlight the factors that predispose patients to increased toxicity from radiation. Esophageal tumors may have different locations, whether cervical or mid- to distal- thoracic, and several factors are to be considered in predicting toxicity, thus further encouraging a model-based approach.
Reviewer 2 Report
The authors have prepared a review on the management of proton therapy for luminal GI cancers. It is very comprehensive. I would like to recommend its acceptance for publication. I only have one suggestion. Please add a space between the value and the unit for all the occasions in the manuscript, i.e., 80.4Gy should be 80.4 Gy.
Author Response
Thank you for your comments. Kindly find below a point-by-point response.
Point 1: Please add a space between the value and the unit for all the occasions in the manuscript, i.e., 80.4Gy should be 80.4 Gy.
Response 1: Suggested edit was done.